# The Role of Mathematics Teacher Education in Overcoming Narrow Neocolonial Views of Mathematics

**Kay Owens**

School of Education, Charles Sturt University, Bathurst, NSW 2795, Australia; kowens@csu.edu.au

**Abstract:** Over the past 30 years, teacher education has changed to incorporate a larger emphasis on understanding students' sociocultural backgrounds, knowing that these influence their learning. However, in terms of mathematics and mathematics education in teacher education, less has been done to recognise the sociocultural mathematics backgrounds of students. An example is provided to show how entrenched colonial attitudes to mathematics have developed into neocolonial policies that influence mathematics education. This example is based on a large historic research project in Papua New Guinea (PNG) that aimed to document and analyse the nature of mathematics education from tens of thousands of years ago to the present. Data sources varied from records of first contact and later records, archaeology, oral histories, language analyses, lived experiences, memoirs, government documents, field studies, and previous research especially doctoral studies. The impacts of colonisation, post-colonial aid and globalisation on mathematics education have been analysed, establishing an understanding of the current status of mathematics education as neocolonial. Neocolonial education policies diminish cultural ways of thinking. Thus, teacher education has an important role in sensitizing preservice and inservice teachers to the impact of neocolonial approaches as well as in developing with students some ways of reducing this impact and encouraging more holistic, culturally relevant mathematics education.

**Keywords:** neocolonialism; ethnomathematics; language and mathematics; postcolonial education; Papua New Guinea; Asia-Pacific





## 1. Introduction

Mathematics education in colonised countries tends to be imported from dominant, overseas countries, especially those that colonised them or subsequently provided significant aid. However, to understand this impact more fully, it is important to understand the mathematics that existed prior to colonisation which is still practised today in these countries, and this mathematics' historical collision with colonialism and neocolonialism. A case study of Papua New Guinea is provided to explore this impact on mathematics education.

Papua New Guinean societies existed from at least 40,000 years ago with several migrations from the north or west. They adapted to various changes such as the minor Ice Age and volcanic eruptions. There are several Papuan or Non-Austronesian language Families with hundreds of languages and a few Isolates (see Table 1). Around 5000 years ago, a major new wave of migration occurred and the Austronesian Oceanic languages developed starting in East New Britain and spreading around the coast and to Island Melanesia as far as Fiji [1]. Groups were relatively autonomous, managing to meet their needs through trade arrangements and intermarrying relationships. There was no central government. The 850 PNG cultures and languages were not influenced by Europe or the Middle East until the 1800s. All these groups developed different forms of technology and economies that required mathematics. This is called ethnomathematics, which varies with each cultural group and language. The environment and ecology influenced its

development so there may be some similarities between some groups. Furthermore, they also exchanged knowledge [2].

**Table 1.** A summary of collected data on counting systems in Papua New Guinea showing the different types of systems described in terms of the cycles of frame words from which higher numbers are formed.

| Austronesian Oceanic | Types | West Papuan | East Papuan | Torricelli | Sepik-Ramu | Trans New Guinea | Minor Phyla | Total |
|---|---|---|---|---|---|---|---|---|
| 2 | **(2)** | 0 | 0 | 0 | 3 | 39 | 0 | 42 |
| 18 | **(2, 5)** | 0 | 1 | 16 | 5 | 86 | 1 | 109 |
|  | **(2, 3, 5)** | 0 | 1 | 3 | 5 | 17 | 1 | 27 |
| 12 | **(2, 4, 5)** | 0 | 0 | 5 | 3 | 31 | 1 | 40 |
| 34 | **(5, 20)** | 0 | 1 | 2 | 17 | 52 | 7 | 79 |
| 4 | **(4), (4,8)** | 0 | 0 | 0 | 1 | 6 | 2 | 9 |
|  | **(6)** | 0 | 0 | 0 | 0 | 5 | 0 | 5 |
|  | **Body-Parts** | 0 | 0 | 0 | 8 | 58 | 4? | 70 |
| 45 | **(5, 10)** | 2 | 12 | 0 | 3 | 4 | 0 | 22 |
| 19 | **(5, 10, 20)** | 5 | 0 | 0 | 0 | 4 | 3 | 13 |
| 73 | **(10)** | 1 | 8 | 0 | 1 | 2 | 0 | 13 |
| 3 | **(10, 20)** | 2 | 0 | 0 | 0 | 1 | 0 | 3 |

After colonisation by Germany in the north and England in the south in the 1880s, in 1902, Papua in the south and, after World War I, New Guinea in the north became Territories of Australia and hence an Australian colony. The funding for the colonies was very limited and so there was little money for education. Australia itself was a colony and for whom, like other colonies, this study has some relevance. Papua New Guinea (PNG) became independent in 1975 so the period of colonisation was relatively short compared to that of many other countries, allowing people to maintain and adapt their cultures. In a sense, its colonisation was condensed in time but had features similar to other places as well as unique features.

## 2. Research Aims and Methodology

The purpose of this research was to document and analyse the development of aspects of mathematics and mathematics education in Papua New Guinea from the past to the present. There are a couple of available bibliographies of education covering colonial times until the mid-1970s [3,4] but these do not focus on mathematics or mathematics education during this period nor from the time before European contact. Despite ongoing research within the country, there has been little on mathematics education per se after the mid-1980s when the Mathematics Education Centre declined [5]. Two exceptions in the 1990s were the doctoral studies of Kaleva [6] and Kari [7] which led to research through the Glen Lean Ethnomathematics Centre from 2000 to 2016 [8,9].

### 2.1. Data Sources

This historical research involved the extensive use of first contact and later documents and memoirs; archaeological and linguistic research from diverse areas and language groups; oral histories; lived experiences; field visits to villages; large research studies on number systems [10], measurement practices [11,12], and mathematical words in different cultures across the country; research studies on mathematics education [5] and teacher education [13,14]; government documents, especially major reports [15–17] and plans recommending changes [18–22] to education; syllabuses; and studies on the language of instruction [23–25].

Many documents, such as first contact documents and linguistic data, did not focus on mathematics per se but it was possible to connect many of these accounts to lived experiences over the past 50 years and students' reports on the ethnomathematics of their cultural

groups to develop significant themes. The mathematicians and mathematics educators who carried out ethnomathematics research shared and facilitated this research consisted mainly of Papua New Guineans from different tribal groups. While colonised views of mathematics were a starting point, there was a general consensus among these researchers that the mathematical practices of their communities, that is their ethnomathematics, were describable, decomposable and able to be re-assembled as mathematics. Technical and ethnomathematical methodologies were important to these communities, embedded in cultural practices and relationships, and passed on from generation to generation. For these researchers and for the UoG teachers who went to their Elders for their research projects, their cultural identity was significant to their professional identity.

### 2.2. Themes and Key Findings from the Data

A grounded-theory approach was taken for establishing themes in this research. The points that kept emerging in the sources provided the main themes, such as the lack of funding or the impact of the use of English as a language of instruction. Other themes were the critical points made by Papua New Guinean education leaders in their reports, such as the importance of maintaining culture while the mathematics education researchers valued their cultural mathematics and noted that the mathematics taught in schools violated [26] their cultural understandings. The themes that emerged from these sources included the following:

1. The languages of mathematics in villages and in schools;
2. The use of visuospatial reasoning in mathematical thinking;
3. The valuing of both traditional mathematics for one's everyday life (once identified) and school mathematics for the dream of a job;
4. The dissonance of mathematics at home and at school.

However, as a historical study of mathematics education [27], there was an argument emerging regarding the impact of colonialism which resulted in the hegemony of educational practices for Papua New Guineans who had received an education from teachers, usually Australian, whose first language was English and well-educated, articulate, high-achieving Papua New Guineans who often received their education from English-speaking teachers. In addition, overseas aid advisers continued to recommend global trends in education from national outcomes-based education to standards-based assessments. The whole education system was affected by these trends. In mathematics education, there was an emphasis on problem solving and a standard, linear approach to mathematical topics which emanated from western curricula. However, when the teachers were school students, often due to a lack of books and equipment, they practised the rote learning of western mathematics and, due to a fear of punishment, failure and letting down their family, they learnt not to speak in their home languages and to follow the (usually male) dominating voice.

Hence, the key findings were as follows:

5. The depth and diversity of foundational/traditional mathematics learning;
6. The growth and sources of neocolonialism;
7. The limitations of neocolonialism;
8. Examples for overcoming neocolonialism.

## 3. Results

### 3.1. Languages of Mathematics in Villages and in Schools

When Lean began to collect counting words in 1968 from tertiary students and teachers who at the time were fluent speakers of their home languages (there was already evidence of these languages changing rapidly), he realised he also needed to carry out village fieldwork and to search worldwide written resources, such as European Enlightenment and Royal Anthropological Institute documents, British New Guinea and Australian Papua annual reports, German reports, and documents by missionaries, linguists and translators. He

took 22 years to complete this huge task. Besides the common lingua franca Tok Pisin, he learnt Tolai (his adopted family's language) and had some familiarity with other languages. It became clear that there were counting words or ways of identifying numbers in the languages of PNG that were different from Indo-European counting systems. He was also able to identify how different systems may have developed and how some languages had influenced others. To do this, he undertook the intricate task of organising these counting systems and comparing neighbouring systems (see Table 1). As a result, along with archaeological linguistic research, which identified proto-languages, he was able to indicate that the Papuan Non-Austronesian mathematical systems had developed and existed for tens of thousands of years and the Oceanic systems for five thousands of years, as well as how they spread (not from the Middle East) and changed [10,28,29].

In my studies with Kaleva on measurement, it was found that there are many words for and grammatical ways to express length, area and volume as well as ways to express forces, comparisons and units of measurement [11,12,30]. However, school policies discouraged the use of home languages so often that there was limited understanding of concepts that belonged to village experiences. The rote learning of western mathematics using English words prevailed, sometimes with little meaning for the learner.

More recently, Bino, Muke, Sondo, Kravia, Sakopa, Edmonds-Wathen and I encouraged teachers to express mathematical concepts in their own language and found that this requires some discussion [31–34]. Nevertheless, most school concepts can be discussed or indicated in cultural ways. However, one of the major concerns has been the failure of teachers, students and communities to recognise the intrinsic mathematical ways of thinking culturally and to consider any mathematics in community activities as quite separate to what they see as mathematics, that is what they learn in school. Our research indicated that in fact mathematical thinking is constantly used in everyday activities [35–37].

### 3.2. The Use of Visuospatial Reasoning in Mathematical Thinking

When Alan Bishop visited the PNG University of Technology where Lean and I worked, he found that the tertiary students who had virtually no picture books (and no TVs or photographs) had difficulty interpreting images of objects. However, with minimal training on how to read these images, the students proved to be very competent. At that point, he decided that there were two distinct capabilities: visualisation and interpreting visual representations [38,39]. Lean and Clements [40] continued this work on spatial abilities with 3D objects and so, along with a number of other studies carried out in this fascinating area, I had a strong foundation for my research [41], culminating in a book [42]. Constantly I experienced villagers making decisions and students and people telling me they were doing it "by eye" or 'in their heads'. Sometimes they used objects such as ropes or steps to explain what they were visualizing. This was affecting all areas of mathematics from their understanding of number size to measurement practices, shapes, geometry, trigonometry and other ratios. For example, in making a house smaller, the Elders were able to visually decide on the horizontal and hypotenuse lengths for a house to keep the same angle. The equidistance of points from other points and points in a straight line were also managed by eye.

Some knowledge and ways of thinking were embodied. For example, parallel lines of a trapezium were understood when walking equidistant from each other between two non-perpendicular lines. The angle of equilateral triangles and the tessellation of these triangles were embedded in visuospatial imagery when planting trees at the vertices using two equal-length sticks. Diagonals were checked for equality when rectangular walls were marked out. A man dragging his foot with a taut rope tied to his ankle and the other end to a stick at the centre point would mark out a circle. A rope with two knots (separating 3, 4 and 5 units or 1, 1.4 (understood as just less than a half) and 1 units) was used by some villagers for obtaining right angles but usually some men were skilled in accurately determining these by looking and coming to a consensus within the group. Knowing the lengths of areas was sufficient to compare areas of roughly the same shape [12]. Ratios

were used for comparing areas of grass needed for different roof areas or for comparing volumes of pigs or pig fat via their girth, length of body, height from the ground and foot area [11].

### 3.3. Valuing of Home and School Mathematics

Many discussions with mathematicians, mathematics educators and teachers suggested that home mathematics was not initially recognised without learning about the field of ethnomathematics. Nevertheless, when architecture students were making their first designed sculpture out of paper without glue or tape, they called upon cultural imagery, patterns and practices to create their small but beautiful sculptures [43]. Students at the University of Goroka remarked on the mathematical capabilities of their Elders when they were describing the activities that the students were asking about for their reports on cultural mathematics. They were honestly proud of their ancestors' mathematics, "even if they did not call it mathematics" [44].

Students also strived to do well in school mathematics. They knew it was a subject they had to do well in to go further in school. There was fierce competition for positions in Grades 7, 9 and 11 as well as in tertiary institutions. However, much of the mathematics was memorised and learnt by rote. Hence, it was clear that for different reasons, they valued both school and home mathematics. It seemed, however, that when they valued home mathematics, they had a greater sense of pride in knowing about their cultural mathematics and succeeded in school mathematics [44,45]. Their cultural identity influenced their views on mathematics and on themselves as mathematicians.

### 3.4. Dissonance between Home and School Mathematics

Without encouraging students to see their cultural mathematics, the students would simply take the view that there was school mathematics and there was different mathematics at home. Many student teachers regarded the mathematics one had to pass at university or school as not existing within culture or being irrelevant. It was only when students were encouraged to make links deliberately that this dissonance began to break down [44,46].

Professional development sessions with teachers [46] and projects of students at the University of Goroka studying the elective *Mathematics, Language and Culture* have shown the strength of recognising cultural mathematics [44]. The following example is from the teacher Mulock Mulung [47], whose lecturer was Wilfred Kaleva. The images are by Mulock, who became a mathematics teacher educator in 2016. It is based on a traditional way of trapping birds that is used in the village of Hotec as well as two other large villages and smaller surrounding villages in the hinterland behind Salamaua, Morobe Province. This is a diverse geographical area. For protein, they hunt and set traps. There are a variety of traps on the land, in the river, in the sea, and in trees. "There are pig traps, bandicoot traps, wallaby traps, snake traps, bird and cuscus traps". He described the making and using of the bird trap, *lek* in his language "*matec*" (his language words are used unless specified as Tok Pisin, the main lingua franca of PNG). It is a special bilum (Tok Pisin) (a loose, continuous string net with each stitch individually interlocking in a figure of eight). "It requires great skill to make it and is carried out by specialists. . . . Catching the birds is a very dangerous activity in terms of men's lives being at risk because they climb and stay in tall trees more than 50–80 m high just to set the trap" and keep a close watch on the birds coming to feed.

"The *habiyom* birds (black with red eyes) come in large flocks from May to September to eat the fruit of the trees used to make canoes". Mulung describes and illustrates the process of making the net. "Ropes are extracted from the bark of *akek* which is similar to the *tulip* (Tok Pisin) tree. . . . Once the bark is removed, it is dried in the sun" and then the fibres are carefully removed. "These are twisted into strong rope". Two sticks are set into the ground so they are 3 m high. The Y at the top of each stick holds a bamboo pole from which a metre-long loop is made and strengthened from which a net with a slight bag is made. The men latch wood to the tree for steps and select a sturdy, wide branch to rest on

while waiting for the birds. (See Figure 1 for each of these steps). "Hundreds of birds are caught over several days and the women come and bring food and collect the birds".

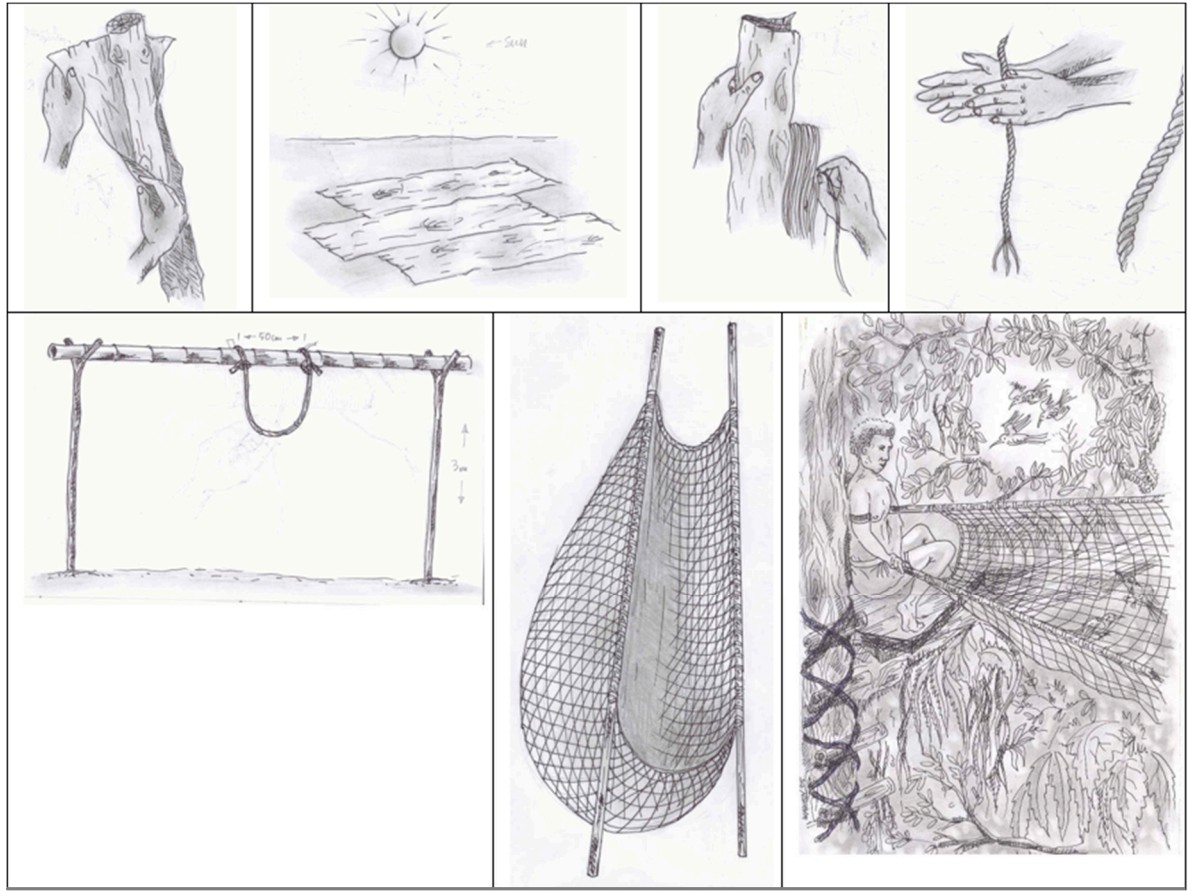

**Figure 1.** Making the bird trap net and capturing birds in a tree. (Drawing and information from Mulock Mutong (2005)).

Using this foundational knowledge, Mulung then prepared a series of examples for teaching the secondary course. He provided background school mathematics in an understandable way before applying traditional mathematics to school mathematics through examples, explanations and exercises related to the topics of area, ratio and rates, and trigonometry.

Steps. The removal of the bark, drying it in the sun, making it into fibres and twisting it into rope. The preparation to make the net and the finished net with the poles. A person who has climbed the wooden steps to the tree branch and waiting for the birds.

"Example. Calculate the trapezoid bed for the net trap (Lek) that has the height of 1.5 m and has lengths 4 m and 2 m respectively. . . .

Example. Those seasonal birds that fly to and fro following their routes fly 80 km in two hours. What is their rate of flight and how far will they fly in 5 h? . . .

Example. Net trap 1 (Lek 1) had caught a total 250 birds in three days and net trap 2 (Lek 2) had caught a total of 750 birds in three days. What is the ratio of birds in three days caught by net (Lek 1) and net (Lek 2).

Solution:

Lek 1: Lek 2 = 250:750

= 25:75 (simplest form)

= 1:3

Ratios are often used to express the composition of a mixture. Ratio of this type can also be used to determine the amount of each component in a quality of a mixture.

Example. A particular pot can hold 24 cups of birds (1 cup = 1 bird), 9 cups of pure water and 3 cups of Gravox chicken curry powder.

(a) What is the ratio of birds: pure water: Gravox (curry powder)?

(b) What quantity of each would require to make 5.0 m$^3$ of birds' soup. . . .

Example. The tree that people climb to set their net trap (Lek) is 80 m tall. The spot where women come to get the trapped birds in exchange for the food - from their men is creating 38 m with the top of the tree. Calculate the distance (Y) from the exchange spot to the base of the tree."

Through this project, this student has shown pride in his relatives' and neighbours' capabilities, strength, perseverance and courage. He also recognises that these are mathematical activities requiring mental mathematical capabilities that can be linked to school mathematics. In some respects, mathematics is associated with the physical and social environment familiar to the students in the same way that a mathematics trail or project is prepared for students to encourage their interest in mathematics by making it relevant to their everyday lives. Further examples are available in other references [24,36,37,42,48,49]. Developing or using such examples is an important aspect of mathematics teacher education if teachers are to provide examples and exercises relevant to students [50].

## 4. Discussion

While these themes were evident in studying the documents and other data, there were more profound considerations emanating from the data sources. All pointed to the fact that there has been and, in most cases, there still is evidence of scientific thinking, technological thinking and the necessary associated mathematical thinking in the various cultures of PNG. However, history has shown that these have not been encouraged in the school curriculum even when there is the desire for school education that reflects the values of PNG societies. This historical study provides some of the details of this occurrence.

### 4.1. The Depth and Diversity of Foundational/Traditional Mathematics Learning

Papua New Guinean societies used mathematics in technology, trade, social relationships, and understanding natural sciences tens of thousands of years ago. Much of this knowledge is still passed on between generations today using Indigenous ways of learning and teaching [27]. Most foundational mathematics is learnt from older men or women who gather under relational connections to share their knowledge in groups during everyday activities or special traditional activities [36,37].

A few remarks might indicate the extent and depth of this knowledge. (See also [10,36,37,51,52]). Seafarers had fishing and navigation skills, travelling over the horizon to distant places [53,54]. There were trading routes and reciprocity to negotiate with items often passed on to far distant places, crossing many language groups [55]. Kinship patterns were extensive, and again, reciprocity was significant [56,57]. There were tools and processes for carrying, collecting, fishing, agriculture, food and materials preparation, creating, building, playing and celebrating [36]. There were designs and patterns of cultural significance and replication of objects such as canoes [37,42], pots [58,59], drums, baskets, string figures [60], shields, bows and arrows, axes, or house walls and roofs [36,42]. All the details for attaining designs, curves, thicknesses, lengths and strengths of objects were mathematical. There was extensive knowledge related to medicines [61] involving spatial knowledge in recognising plants and where to gather them, and knowledge of how to treat illnesses with different medicines and processes.

The classification and sets of designs were sophisticated and related to culture [51,62]. These are evident for the shapes of objects and on the various parts of canoe boards, house boards [52], shields [36], other carvings, leadership symbols [2], food containers and pots [58,59,63]. Actions, their order and links between them have been studied in string figures [64,65] but also in making other items, such as string bags (bilum) [35,66,67]. They are remembered but also reorganised to create new designs. Patterns occur in gambling practices [68], weaving and making string bags. Numeral systems are varied, with some

being unique and some shared with neighbours. Some are linked to collecting, measuring, trading or classifying [28]. Importantly, counting often has sophisticated systems and cultural importance, as indicated by Owens and Lean [29] and with Paraide and Muke [10] who also provided theses on their own languages [24,69] respectively.

Mathematics teacher education needs to ensure that teachers do not restrict these mathematical concepts to Euclidian geometry or base 10 counting systems. There is a wealth of cultural examples that indeed extend current school curriculum ideas on classification, pattern, design and numeral systems. Furthermore, these mathematical approaches link students spiritually to mathematics.

### 4.2. The Growth and Sources of Neocolonialism

Neocolonialism in PNG is a result of colonial education policies and practices but also the continuing expectations and practices of aid advisers and nationals.

### 4.2.1. Historical Developments

In the late 1800s, a few anthropologists visited PNG (e.g., Mikloucho-Maclay [70]), European sailors navigated its waters [71] and a few German business people began plantations or recruiting for other plantations in the Pacific region [72]. Missionaries soon followed, sharing the gospel of Jesus in the vernacular languages, often in a religious format but also assisting villagers, especially with health issues and education [72–74].

Governments felt the need to set up administration and controls. The German government in the northern mainland and islands soon set up administrative centres, laying claim to it as a colony in 1884. This prompted the British to lay claim to the southern side close to Australia, leaving the colony of Queensland and later Australia to administer Papua. One issue of the early administrators was the exploitation of 'the natives' as they were called. This encouraged them to provide a basic education. Mostly it was through supporting the mission schools but then they began requesting that schooling be in English so that the administrators could converse with the natives. Money was attached. After World War I, the League of Nations passed the northern section to Australia as a Trust Territory. Gold mining was exploited as was already occurring in plantations. This provoked many foot patrols into the virtually unknown, unpacified highland areas which were then opened up since aircraft were able to fly there. Importantly, this impacted the local economy as quantities of kina shell money were imported to pay the workers from the areas and encouraged people to travel to other people's land for work. Using pounds (made of paper like a 'leaf') and shillings was suitable to the digit tally (5, 20) cycle counting systems of many. These terms and translations for this 'money' continue to today (field visit to Malalamai, 2006) [75].

In the Australian Territories, English was stipulated as the language of instruction for government funding [4]. Not only the locals but also the Germans were required to have schooling in English. However, overall, little money was available to support the two Territories' colonial administrations [76,77]. Already the dominance of English as the valued language had begun. Teachers now need to make an informed and concerted effort to use a local language and not slip into a lingua franca such as Tok Pisin. However, too little has been done to explain basic mathematical concepts, such as arithmetic operations, in terms of local languages. These were rote learnt but widely used in employment.

Interestingly, in early British New Guinea and Papua administrative reports, basic word lists of the local languages were recorded as new centres were set up. However, most of the language work was carried out by churches. In Port Moresby, Lawes [78] and colleagues had written down the Motuan language by 1885 and used it in large schools for the local people [4]. Other village languages were also used, especially Dobu in the Papuan islands, Tolai in East New Britain, Bel in Madang area, and Kôte and Yambim in Morobe and beyond for churches and schools [73]. Students completing the two levels of the basic curriculum or later Grade 6 would be recruited as teacher assistants in schools. South Sea islanders also came as pastors and teachers [4].

During and after World War II, there was one administration [3] but no national curriculum so teachers taught what they or their senior teacher knew as mathematics from their home countries. In the 1960s, the Australian Prime Minister started to talk about autonomy [79] for the Territories. Around the world, more and more colonies were becoming independent. However, education in PNG was very limited and insufficient for autonomy, let alone for an independent country. Hurriedly, high schools were set up. By then, many Australians, often quite young, were recruited as kiaps (administrators in charge of areas) and teachers to remote areas as well as towns and coastal centres. Teachers' colleges trained both Papua New Guineans and Australians [80,81]. Some students were selected for studies in Australian secondary schools and universities (as per personal communications with such students between 1970 and 2016). By 1966, the University of PNG was set up in Port Moresby, and the beginning of the PNG University of Technology was not long after [82]. There were graduates by self-government in 1973 which preceded Independence in 1975. Research into education, particularly mathematics education, was strong and began influencing worldwide research [5,83]. Teacher educators and senior high school teachers were mainly from overseas but, for a decade, PNG teacher educators were trained as a group for two years in Australia [13]. Following that, there were Australian and New Zealand Awards for Masters degrees, in-country Masters, and more recently a couple of intensive courses in PNG.

Further details can be found in the bibliographies previously mentioned [3,4], Paraide et al.'s book [27], an earlier summary by Owens et al. [84], and another long-term education researcher, Weeks [85].

### 4.2.2. Colonial Impact on Education and Languages

The administration of the colonising countries focussed on law and order, taxes, and keeping records of businesses and other groups such as churches [85]. Initially in the early and mid-1900s, funds went to government schools and to missions if English was the language of instruction and were proportional to students' achievement in English and mathematics examinations set by Queensland (an Australian State). Missions or churches dominated education training and still do today; all but one of the primary teachers' colleges are run by churches ([13], see Appendix of 27) with the Institute of Education mostly concerned with early childhood education. The University of Goroka also provides Certificates and Degrees in Early Childhood Education and degrees (including Masters) in education for all sectors [5].

Before self-government was set up, Australia instigated a 6-month training program in Rabaul, mainly for Australians. In following years, ASOPA in Sydney provided some understanding of cultural diversity and respect for students undertaking school education, certificates and degrees [86] for teachers going to PNG. Before and after Independence, there were committees to advise on curricula for primary and secondary education and teacher education. The college staff members were able to be in touch and share their ideas and strengths. Some overseas mission staff members were in the country for many years while others came for short terms [5,13]. Recognising the needs of village children in primary teacher education was evident, and it was partially reflected in the official use of the term Community Schools after Independence. Nevertheless, the local PNG teachers tended to think schooling was the way they were taught by the Australians, and for mathematics, this involved considerable use of rote learning, although Dienes influenced a number of schools [81,87,88] with the idea of mathematics as logic and the use of games with apparatus for teaching. Like many good ideas, most of the materials sat idle in school storerooms up to the 1980s as there was not sufficient professional development for teachers. This coincided with an increasing number of Grade 10 students completing two years of teacher education, and by this time, expatriates were not expected to hold primary school teaching positions (International Schools fell under the government's International School Agency which continues today. Students and teachers are Papua New Guinean).

There have been schools using local languages for teaching, e.g., Tolai in East New Britain, Tok Ples or local church languages in remote Morobe, Enga, Milne Bay and Bougainville [89,90]. However, school students from 1960 to 1985 reported they were punished for speaking languages other than English in both government and mission schools [23]. Since the education system meant that students often left their village for a small centre, they were already beginning to use a non-home language. The children then went to high school, senior high school, teachers or other college or university where English and Tok Pisin were the main languages between students. After years of education away from their village, teachers might or might not go back to their village area to teach. Many students struggled to keep their culture and vernacular language and to acquire their family's foundational knowledge. Despite this, they still had strong connections and pride in their family and their family's foundational technological and mathematical knowledge [43,44]. Was the loss irrevocable?

### 4.2.3. An Indigenous Voice

Before Independence, a committee of educated Papua New Guineans chaired by Alkan Tololo prepared a report for the Department of Education [15,91]. They recognised the importance of students valuing their culture, knowing how to live in their villages, and connecting village knowledge and school knowledge. The second version was more nationalistic in presenting a philosophy recognizing cultures and languages and aiming to preserve the numerous societies of PNG through education [92]. However, there was still an Australian responsible for the Territories, and he could not see how this report could be implemented so he went to the expatriate Dean of Education at the University of Papua New Guinea who hurriedly prepared another education plan [84,92]. There was perhaps some concern that the capable and elite Papua New Guineans should have the opportunity for a western education without the expenses of international schooling [93]. The Australian curriculum schools were replaced by International Agency Schools. These schools enrolled expatriate children, mixed-race children, and the children of professional and business Papua New Guineans.

By this stage, it was recognised that different cultures counted by different cycles and were not all base 10 systems. Since the late 1970s, teachers' colleges encouraged their student teachers to learn basic words of the language of their students if practicing in a village school. This included the counting system, arithmetic operational words and ways of measuring. The Mathematics Education Centre at the PNG University of Technology, the Education Research Unit at the University of Papua New Guinea and the Department of Education drew together many mathematics research studies, resulting in a special issue of the *Journal of PNG Education*, Indigenous Mathematics Project in 1979 [5]. The textbook for secondary schools was called *Mathematics Our Way*. The team leader was a New Zealander, the team members were highly committed PNG curriculum writers. Then the expatriate Oxford Press came with textbooks and teacher guides. Such glossy materials did not last in schools, and curriculum changes were made, perhaps trying to raise standards.

The idea of a preschool education in vernacular languages, while successful in many places, was not supported in 1975 or over the next decade, although Provincial leaders advocated for it and other ideas about implementation. This was perhaps the only value in internal assessment comparisons which, as Weeks [84] pointed out, were almost an obsession with several detrimental effects when tied to a lack of funding, such as no increases in secondary enrolments and the rise of de facto secondary schools (distance education, as well as vocational and technical colleges). There were no new senior high schools as planned and the monies taken from the universities did not reach the schooling sector to increase enrolments at any level. This did not help students who were unable to find employment or the urban drift as some might have thought [84].

The opportunity to hear and develop the Indigenous voice was lost at this stage and indeed for 10 years until 1986 when another Indigenous committee, this time chaired by Paulius Matane, wrote a report for which plans were made [16]. The World Bank continued

their financial support and educational reports. They supported the idea of achieving universal primary education through village schooling. Thus, cultures and languages were recognised in schools. There was an opportunity to incorporate village mathematics.

### 4.2.4. Attempts to Educate following the Indigenous Voice

The government accepted and began implementing the Matane report and the Reform period began [94], albeit rather slowly and inconsistently. The whole structure of education was to change as well as the curriculum. To have village schools, the community needed to provide the school and the teachers' houses. Teachers who knew the language and had achieved a Grade 10 could be recruited. PNG was attempting their own education and not just following externally dominated ideas. The structure of education was changed to three years in elementary schools (Pre-elementary, Elementary 1 and 2), six years in primary schools (Grades 3 to 8), and four years in secondary schools (Grades 9 to 12). The desire for universal education meant elementary schools in villages would use the home language of the children [95,96]. However, setting up this system especially in remote areas was problematic although changes were gradually made [97,98].

There were more PNG educators with higher degrees, and curriculum advisory committees had strong national representation from practicing fields, universities and schools. They set high standards for mathematics and teacher education within the constraints of time. However, they were not necessarily meeting regularly as they were before 1990 to share ideas [13]. In essence, it took about 20 years to implement this change but still there was insufficient teacher education.

### 4.2.5. Funding Affecting Teacher Education, Research and Materials

Funding was an issue. It was taken away from higher education. The maintenance of higher education institutions and research could not really continue as before, and even getting government funds for salaries was problematic (T. Chan, personal communication, 1997). However, the money did not reach the school sector in terms of implementing the Reform curriculum. There was no funding to assist with the necessary input from Elders into the languages of the schools.

To survive, elementary teachers worked for half a day, so they could tend to their gardens in the afternoon. Teachers first trained under the head teacher and were accredited upon inspection if they knew the local language, had a Grade 10 education and had undertaken training. Then they would be paid a full salary. However, for years, training was often not available and inspectors found it difficult to visit. Many teachers received no or inadequate salaries.

An Australian advisory team, whom it was said had too much say, was involved in developing the system of teacher education for elementary schools. The teacher education courses were set up as Self-Instruction Units with a short introductory workshop, often given as lectures to a large number of teachers in a village area. At first, teacher education was delivered by travelling Institute staff members and then by Provincial Education Officers with varying skills, training and experience. The motivation of teachers varied considerably (personal communication, T. Hamadi, lecturer from PNG Institute of Education, 1997). There was not a full unit on teaching bilingually and transitioning from the vernacular language to the English language and there was not a mathematics unit developed by the Institute using cultural mathematics. Teacher educators and teachers were not sure of how to establish cultural mathematics.

A lack of funding led to loans requiring repayments by the government and more overseas aid with more overseas advisers with their own (neo)colonial views.

### 4.2.6. Curriculum Changes

There were now two Australian-funded projects for curriculum changes. Both had highly committed expatriates, mostly Australians, and each had a PNG counterpart. One project was the Primary and Secondary Teacher Education Project. This developed new

curricula for colleges and encouraged interactive learning, the use of computers for knowledge (e.g., mathematics textbooks and encyclopaedia) and some communication between lecturers (at least principals), and the implementation of gender equality (note the idea of equity was not even considered). There were certainly improvements in teacher education, and some senior staff members undertook Master's degrees. Nevertheless, evaluations suggested that, on the main issues of bilingual education and gender equality, there was still a long way to go and that there was still no compulsory or apparently taught cultural mathematics subject in primary teachers' colleges. Bilingual education was not even considered by mathematics curriculum lecturers [23,99]. Ten years later, both the language and the mathematics subjects were regarded as too difficult for the lecturers replacing those who were able to take advantage of the PASTEP project (personal communication, Jones, UK Volunteer Services Overseas (VSO) team leader, 2015).

It took 10 years after Tololo's committee first emphasised the importance of universal education suitable for students in their ecological environment before the committee chaired by Matane reiterated the same values. Syllabuses were beginning to appear in the early 1990s, but it was not until another aid project began, 25 years after Tololo's report, 23 years after Independence, that a concerted effort began to bring language and culture into the curriculum, especially in elementary schools. However, many issues were not adequately addressed by education authorities or aid organisations [84,100].

The group of Australian Aid advisers took over a year to actually involve PNG Curriculum and Assessment officers in their work. The advisory team from Australian Aid (Curriculum Reform Implementation Project) introduced Outcomes-Based Education (OBE), then common around the world, but the directive required short syllabuses. These proved to be inadequate, and after the elementary school level, there was no initial or strong Indigenous voice in the mathematics curriculum documents. The Teachers' Guides and expensive textbooks that were essential for implementation soon disappeared, just as the earlier books had disappeared. OBE began to be seen as the problem for education by the elite and others. The country was in a dilemma with its lack of funding and new neocolonial curriculum. Like many of the reforms in mathematics education, even going back to the introduction of Dienes blocks, it was inadequately supported by teacher education or inservicing [6]. Teachers and educators wanted their culture involved but could not see its implementation in the curriculum.

### 4.2.7. The End of Learning Cultural Mathematics in Home Language

In 2012, O'Neill was elected as the Prime Minister with the promise that English would be the language of instruction from the start. This was promised even though so much research supports learning mathematical and other concepts in one's home language and bridging them later into English as the best educational approach, although students were not doing well on Pacific standardised tests [27]. The elementary schools disappeared and were replaced by early childhood education centres for two years (having a play-based first year and picking up the pre-elementary syllabuses from the elementary schools), and then the students had to go to primary school for Grades 1 to 6. Mathematics was no longer called Cultural Mathematics. There were restructures, yet again, of the education school system [101]. In fact, instead of Australian colonialism, Japanese approaches to mathematics began. The English version of a Japanese textbook was now available for teachers to buy if they did not receive it from the Department of Education. Standards based assessments, following world trends again, were introduced.

### 4.3. Overcoming the Limitations of Neocolonialism

The Matane report was an attempt to overcome PNG's colonial legacy, and this Reform era had high potential. The issue of disappearing funds and the need for more overseas aid to implement changes made the process problematic. Nevertheless, significant changes were started.

The language issue for PNG was significant. Language and culture are closely interconnected. At the time of Independence, with 850 languages, most of which were still the children's home language, English was rarely heard in villages and homes. Since Independence, the lingua franca Tok Pisin began replacing home languages rather than English. The loss of language and of understanding what was being discussed in the classroom if English was the language of instruction was exacerbated by the amount of time students were studying away from home, even from the beginning of school. Due to contact with people from other language groups through schooling and migration to the towns, English and Tok Pisin were taking their toll on local languages and cultural practices, especially those that required young men to carry out the tasks. With the Reform, elementary schools were available in the village or nearby village and they required teachers to have the home language of the majority of students. There was mathematical terminology, especially counting, to be heard in the classrooms. There were few or no resources to assist the teacher and minimal teacher education. However, some of the earlier generation of school children had not heard these words and were excited and proud to hear them in the classroom (personal communications, 1999 to 2003). However, there was no clear implementation process for the policy of transitioning (bridging) from vernacular to English which was to occur towards the end of Grade 2 (the third year of school) and the first year of primary school (Grade 3) and continue throughout later schooling. Nevertheless, there were some good ideas such as the use of 'shell' books that told in pictures a probable village story and on which the local language could be written. These were used to read with the class. The schools that had the support of SIL volunteers (Summer Institute of Linguistics volunteers who were primarily recording the local languages and assisting with the translation of the Bible) were doing well in terms of using phonics for bilingual transitions, teaching materials, and mathematical language but there were still many languages without an agreed orthography. At least 400 languages were being used in schools.

Money was needed for all the language tasks but also for developing each culture's mathematical ways of thinking and discussing mathematical concepts. These tasks involve Elders who were already busy surviving in their rural environment, and it would take time, support and money to discuss and establish cultural mathematics and language. There were no government or aid projects implemented for this. Some teachers were able to bring local language into mathematics besides counting but this was a high expectation without considerable support. The lack of teacher education for elementary schools and the lack of resources meant that students were not adequately learning to read in Tok Ples or English and their mathematics was just as poor.

Educated Papua New Guineans, such as O'Neal, who had opportunities in Australia, still felt that the only good education was one that met overseas levels of education, including in terms of the curriculum and language of instruction. At first, when the standards were introduced, some thought that this meant English had to be used as the standard across the nation with the same lessons taught to every child. An effort was made to educate the advisers who educated senior teachers who were then to inform teachers about the standards. The Assessment and Evaluation Division of the Department of Education was informed of our discussions, and they began to realise that the standards were ways of measuring the achievements of outcomes. Early childhood teachers were already orally monitoring their students during classes. In reaching for the highest standards for their country and avoiding the stigma that Australia now had because of its connection with OBE, the country "looked north" (a former Chief Minister's slogan) to Japan.

The draft of the elementary curriculum was too hard for PNG elementary teachers to follow, and a teachers' guide was prepared based on earlier SIL materials with so-called scripted lessons. There was little connection between the two documents but gradually the syllabus was reduced, and a Japanese-based English textbook was made available to teachers. There were still no links to cultural mathematics. For example, there are many examples of line symmetry and rotational symmetry in PNG cultural artefacts and practices, but the textbook example was on tiles that are probably only found in exclusive hotels and

are unknown to village children. Even though OBE and the use of local languages were being blamed for the students' poor level of reading, which was seen as an extension of colonialism, there appears, ironically, to have been even less emphasis and training on cultural mathematics models of concepts in the PNG curriculum than in that carried out within Australian Aid projects. Ethnomathematics was not considered and local language learning was discouraged, although teachers would use it in an ad hoc way if it assisted students to make sense of a concept [25]. Furthermore, in primary and secondary schools, external examinations still mattered for the selection of students for the next phase of education. Assessment tasks and criteria were, however, suggested in the syllabus and textbook for improving students' learning while teaching. A global mathematics system could be identified.

### 4.4. Examples of Overcoming the Limitations of Neocolonialism

The Reform attempt to introduce a PNG education with goals set by PNGians for PNGians was dismissed by the government due to a lack of vision for the need to support language work at the grassroots level, a lack of knowledge in overseas aid projects on local languages and mathematics, and growing neocolonial attitudes within the country. Nevertheless, a number of projects show what might be possible to overcome neocolonialism.

#### 4.4.1. Teacher Education Units on Ethnomathematics

From 1990 until 2016, several lecturers supported the popular elective subject for teachers, *Mathematics, Language and Culture*, at the University of Goroka. These included Wilfred Kaleva, Rex Matang [102,103] and Charly Muke, who all had relevant Master and/or doctoral degrees. In 1996, they were supported by the American Richard Zepp and in the 2000s by the Australian Kay Owens. The students prepared research reports on the mathematics of their and/or another's PNG culture. They then made links to the PNG curriculum, usually the high school curriculum.

Over 230 of these reports were analysed for reference to mathematical measurement ideas, but in doing this, it was also evident that traditionally, people made use of visuospatial reasoning (see Section 4.4.3). Students were able to identify many areas of the mathematics syllabus (usually those of secondary schools) which related to their village activities and/or artefacts. These especially included designs and orders of steps with links to algebra; measurements, especially length, volume and angle; trigonometry; and geometry. Importantly, students were proud of their ancestors' mathematical thinking and capabilities even if they did not call it mathematics. From this sense of identity, students were appreciating how mathematics could relate to their community life, encouraging their mathematical identity [42,44,104]. This approach to professional identity through cultural identity was also evident in the aforementioned project by architectural students (see Section 3.3). The students proudly called on their cultural backgrounds to develop design, joints, balance and problem-solving skills [29].

#### 4.4.2. Early Childhood and Early School Self-Instruction Unit

A research team (2014–2016) funded under the Australian Research Development Awards developed a Self-Instruction Unit on mathematics teaching and learning that was given to the Institute of Education [105]. The materials included a comprehensive model of teaching that incorporated culture and language, mathematics and early childhood mathematics education. It was very practical. It was accompanied by small books based on activities to be found in villages in PNG on concepts such as composite numbers, measurement of area and number patterns, which lead to the concept of multiplication. The pages could be translated into local languages. There were also videos of cultural mathematical activities, classroom games, and how to use the early mathematics assessment tasks based on Matang's work [102,103] and practiced in workshops. Teachers who joined in the remote workshops valued [106] what they learnt, but the workshops were only 3 to 5 days. This was too little, too late. SIL was beginning to make good inroads into teaching

teachers how to teach bilingually and to recognise cultural mathematics (at least their counting systems) when the Reform was stopped.

The materials [105] were also valued by teachers in the Solomon Islands and Tonga, but again funding would be needed to implement the ideas more widely. There was evidence, however, from the Madang Province that the information that was given on computers in the later workshops, rather than that in hard copies, was not being utilised fully. However, the main issue was the difficulty of bringing about change when political changes counter the purpose of using local languages by making English the language of instruction and by changing the curricula.

After years of English (or Tok Pisin) education in the country, it is now difficult to arrest the loss of languages or the devaluing of home languages in education. The loss of cultural mathematics and its use in understanding school mathematics is also evident. Teachers did not have access to national or international research on the strengths of learning and understanding mathematics in home languages or learning in multilingual situations [23,31,33,107–110]. Expertise was not readily available for ongoing professional development. Teachers needed more support for teaching in their home language and transitioning to English. Nevertheless, Australian First Nations are reviving their languages which had been often considered lost due to Australian protectionist and assimilation policies. Perhaps it is not too late for PNG.

### 4.4.3. Recognising Visuospatial Reasoning as a Key of PNG Mathematical Thinking

Voices such those of Charly Muke and Patricia Paraide on learning in one's home language, teaching bilingually and transitioning to English were being drowned out. However, in 2022, Charly Muke, a plenary speaker at the International Conference on Ethnomathematics 7 (ICEm-7) (hosted online by PNG as well as other countries), said that teachers and administrators now need to do something differently because they were stuck with English. If English is decreed the language of instruction from early childhood onwards, then there need to be alternative ways forward. Muke noted how he sat in primary school not understanding a word but for mathematics, with concrete materials, he figured out what was going on in his own language in his head. Perhaps, said Muke, we need to consider how Papua New Guineans think mathematically when they are doing cultural activities that often involve science, technology, engineering and mathematics.

Firstly, we know they think visuospatially in these contexts. They often call it 'in my head' or 'by eye'. How do they do this? Already the work of the secondary school teachers mentioned above could be extended in discussing this way of reasoning mathematically. Owens noticed the use of ratios as mentioned in many examples in Section 3.2 above. The regular use of a bit of rope as a measuring unit to mark equal distances can be adapted to lessons on measurement. It is also used for circumferences, such when one is collecting and flattening out bamboo for floors, or making a decision on the sizes of pigs. Rope tied to a post and the leg of a man dragging his foot as he walks is used to mark out the circumference of a circular house. Its length and the ultimate volume of a house are visuospatially linked. Sticks are also used for measuring lengths as they can assist in equalizing or halving spaces between posts or spacing *morata* (made from sago leaves sewn over narrow planks of limbom palm) for covering a roof. For measuring shell money, a fathom from one's outstretched arms is used. Steps, hand spans, the fist to the elbow (especially for one's girth) and finger parts are commonly used. The height to one's arm pit, shoulder or head is commonly used for heights or parts of houses. Other readily available tools such as spades are used for deciding the depth of trenches and slopes. Estimates can be made of slightly longer and shorter measures by sight.

The physical embodiment of measures also aids visuospatial reasoning, such as using walked lengths and directions, feeling the swells when sailing, knowing the strength needed for bows, and marking the passing of time when doing activities such as sailing, walking, fishing or sleeping.

### 4.4.4. Recognising Ethnomathematics

Ethnomathematics is about mathematical processes. In Section 4.4.3, we discussed visuospatial reasoning which incorporates displays and representations. These draw meaning from another important aspect of ethnomathematics, that of learning mathematics through group work and discussion. In our field work, Owens noticed a discussion of the ratio of two sticks forming the sides of a right-angled triangle so that the slope remained the same when one of the sticks needed to be shorter. When an Elder was showing us how to make a difficult diamond pattern while weaving, another Elder pointed out where an earlier line was wrong, creating the mistake. The Elders were discussing the mathematics involved in weaving. One student teacher noted the large numbers of people who would gather for making some decisions such as bride-price or land distribution [44]. Most decisions involving mathematics at a cultural and community level are discussed, so the encouragement of thoughtful discussions mathematically would help mathematical learning in schools. Our workshops asked teachers to notice who was doing the talking in their classrooms [33]. Following the workshops, we found teachers implementing activities to encourage discussions and inquiries. Group work was not just for practice but to enable a discussion or establish more than one answer to a question.

Using representations for numbers is commonplace. Muke noted that his father marked parts of his body for different decades. Bodytally systems can be found in several western provinces [10] (see Table 1). Pig tusks, shells, bamboo pieces and knots on ropes often mark numbers while leaves may be torn from a palm frond to indicate the passing of days.

Muke also recommended the use of traditional games in teaching. He illustrated how their betting game with stones involved number operations and probability while cat's cradles illustrate the ideas of polynomials, sequences and shapes. He also recommends studying their counting systems and representations (see Section 3.1) [19].

Teacher education for multilingual classes and cultural mathematics needs to be compulsory. Our work [10,36,37,42] provides sufficient examples to be used along with students continuing to provide their own examples to encourage this practice as a matter of course. Muke's [25,111] study is illustrative of how good teachers are likely to use local languages for explanation but much more is needed for strengthening the mathematical register in these languages. Using a transliteration for the word 'multiplication', for example, does not really provide the meaning of the word. However, there are numerous expressions for 'equal groups' that would strengthen its meaning. Likewise, establishing the meaning for 'division' can easily be discussed, for example, in sharing long lengths of shell money with relatives [37,96]. This example also could provide the notion of unequal lengths, since more connected people might receive more shell money, and the notion of ratio. Paraide, who initially learnt mathematics in her vernacular Tolai and was exceptionally good when she started school in English, experienced another issue regarding colonialism in the absence of an emphasis on cultural mathematics. Because her parents did not go to school, her expatriate teachers made her feel uncomfortable at school, and this was exacerbated by not having other students to speak her language to discuss problems in mathematics [27].

During the education Reform sparked by the Matane report to use local languages and cultures, notably, there was little work done on local languages for mathematics outside of the counting words. Lean's work, though available in teachers' colleges, universities and Education Departments, was not being utilised to strengthen an understanding of counting systems, cycles or bases. There was no systematic education on how to record mathematical terminology in a local language. Implementing this would be costly and require many skilled people. Could the Teo Māori experience be repeated even in a small way [112]? A list of mathematical terms for primary school were translated into local languages in workshops by teachers and Elders in discussions but often only a few terms were explored in the short time available [31].

Our research over decades on ethnomathematics in PNG is mostly summarised in our books [8,19,28] and papers, for example, [30,51,113].

## 5. Conclusions

There is a way forward to address Muke's suggestion that schools need to implement mathematical ways of thinking that have a cultural basis. In other words, knowing and valuing ethnomathematics and associated mathematical ways of thinking and learning could be modelled in terms of school mathematics and teaching it [114–116]. Furthermore, teacher education can support these changes and, in the process, change teachers' understanding of mathematics, cultural mathematics, language, the ways of teaching mathematics, and the politics of education in a neocolonial country.

### 5.1. Ethnomathematics and School Mathematics

Modelling requires the recognition of classes and systems often associated with patterns. The various classes or parts have relationships through the systems.

#### 5.1.1. Classifications

There are sophisticated classification systems for counting, design, art (on cultural artifacts), gambling and kinship (see above and cf. [117,118]). Classification in school geometry, for example, is simplified and its relevance reduced by not having a spatial and cultural component. Every language has classifications. Canoe decorations provide one example [51]. In PNG, many counting systems, especially among the Austronesian Oceanic languages and some neighbours, are based on classifications [119].

#### 5.1.2. Space and Geometry

People's knowledge of places and a mental map of large areas are held in their heads as they traverse forests or seas [35,120]. This knowledge involves position but also their visuospatial knowledge of trees, soils, water movement, winds, reefs, fish, sharks, dugongs, shell fish and other creatures that inhabit different areas. The interconnectivity of the mathematical aspects involved, such as position, shape and vectors, has a purpose. Having a purpose is a main driver for learning, remembering, and making connections between mathematical ideas [42].

People's knowledge of complex trade, intercultural relationships and reciprocal agreements [118] involves complex accounting systems covering many goods and money (PNG kinas or traditional money, e.g., shell *tabu*). Pairs, matching, equality and inequality, and increase and decrease are central to these systems. All these are mathematics concepts. Some mathematical knowledge is recorded, often on the body in some way or by objects and displays [10]. Representations include tattoos, body parts, displays, *bilas* (body decorations), the demarcation of land, house sizing and design [24,28,121,122]. All cultures have mathematical thinking for activities—counting, measuring, designing, locating, playing, explaining [83], understanding, interpreting, inventing and reasoning [123]. These are techniques for and models of cultural ways of thinking mathematically.

### 5.2. Implementing Ethnomathematics in Schools

Listening and working with Elders is essential [124–126]. Money is needed for this. First, a range of mathematical activities needs to be discussed and the mathematics needs to be teased out and represented, as in mathematical modelling. The mathematics might not easily fit into the school curriculum but could be used for patterns and relations. For example, string figures show algorithms and inventions, while canoe boards show classifications and patterns. Designs, e.g., *kapa* (round leadership symbols made of hard shell and tortoise shell) have diverse symmetries, patterns and angles. Ways of counting have systems, and many can easily be coded (personal communication, Kari, 2003), while others indicate intricacies related to cultural practices. Each basic counting system can be classified using frame words (the basic words from which others are made), cycles (which

indicate the systems for making high numbers). In most cases in PNG, this is a more appropriate approach than using the term base. Many are digit-tally systems with (2, 5, 20) cycles [127] (see Table 1). Appropriate teacher education is essential to assist teachers to analyse the counting systems of their students and others in PNG in order to make links between systems including base 10 [14].

In addition to the work on foundational/traditional mathematics given in the two chapters of Paraide et al. [36,37], Owens [32] discussed cultural implications for discussing large numbers, groupings, time and work patterns, transactions, classifications, art and design, and Bino [67] indicated mathematical thinking for model canoe building and sailing. More importantly, she showed the significance of this ethnomathematics for social justice in providing a means for equality and money in a rural situation. In cultural practices, people discuss problems and situations that need resolving. They share their conceptual understandings which are generally associated with visuospatial reasoning which is a holistic way of presenting the problem. Concepts, comparisons, memories of the past related to a problem or object, patterns, parts, size and shape are all considered visuospatially and ecoculturally. An environment supports and constrains patterns of activities and the diversity of responses. These sophisticated ways of thinking need to be expounded more by teachers, villagers, researchers and curriculum writers. This idea of mental mathematical thinking which generally includes visuospatial reasoning [28,35] needs to be captured in mathematics and these thinking skills brought to the fore in school mathematics in PNG if neocolonial losses are to be overcome.

*5.3. The Importance of Teacher Education*

The study by Quartermaine [13,88] noted the significance of involving teacher educators in decision making and having regular contact between them for generating quality teacher education. In this way, new approaches to mathematics education could be introduced. A decade later, Tapo [11] who was looking at effective teacher education to implement the recommendations of the Tololo and Matane reports, also recognised the importance of curricula changes and the professional development of teacher educators. There is no doubt that, around the world, quality teacher education is at the heart of quality teaching in schools. With the constant turnover of staff and the poor state of teachers' colleges in terms of their facilities, low salaries and gender equity, as well as opportunities for reading research and carrying out research, there is scope for improving teacher education.

However, it is essential to highlight two areas of the curriculum. First is the need for recognising ethnomathematics. In Australia, all teacher educators undertake an awareness of Indigenous education and must achieve competency in this area. There is a limited voice for ethnomathematics but there is a strong voice for Indigenous education. These include the Aboriginal and Torres Strait Islander Mathematics Network in which mathematics, business and education are connected; Yunkaporta's [128] eight ways adopted by the NSW Department of Education; the international group Indigenizing University Mathematics; the Aboriginal Education Consultative Groups who are keen to promote Indigenous education in general; and the Stronger Smarter Centre which focuses on teachers and teacher education encouraging this in students rather than a deficit approach. There are some parts of the National Curriculum for Mathematics in which First Nations are recognised.

A former Queensland Center (called Yumi Deadly) and Chris Matthews used the Goompi model in which reality is abstracted to mathematics through creativity, symbols and cultural bias, and then this mathematics is reflected upon to create a new reality through the same processes. In Australia, as in PNG, often good work in ethnomathematics has been done through projects that run only when a grant is available. It is important that ethnomathematics is part of the curriculum and teacher education for it to remain influential in mathematics and teaching.

The second aspect of ethnomathematics is mathematics teaching and learning. Morris [129] noted that the Goompi model is also applicable to teaching. In particular, a teacher needs to respond to the cultural background of the students. For example, students may

share their stories to present their reality, and their teacher needs many teaching strategies to be creative in responding to the students and for them to abstract the mathematics. Their reflection, as prominent in Indigenous cultures, also encourages further application to the real world and to mathematics. She notes the following:

> "The Goompi Model provides an excellent framework for teachers to enact this and follow a cycle of learning that takes students from their everyday reality to the world of mathematics and back again by connecting maths with culture. Throughout the whole process, students' cultural backgrounds are supported and reinforced while also seeing themselves as mathematicians for tomorrow" [129], p. 192.

Ethnomathematics has strong research groups in other countries with displaced and disadvantaged communities, such as in Brazil, Peru, the USA, and African countries, especially Mozambique and the Republic of South Africa. Interestingly, both Nepal and Indonesia have ethnomathematical research studies underway. Ethnomathematics has made a considerable difference in Hawaii where it is recognised at the University providing higher degrees in this area.

Ethnomathematics needs to be a compulsory subject of teacher education in all countries where there are First Nations, colonisation and/or multiculturalism resulting from both the way that the country was formed and from immigration. When this became an elective subject in PNG teachers colleges, there were often various time and organizational constraints. Furthermore, ethnomathematics goes a long way towards meeting the PNG goal of universal education which would provide an education for rural communities without access to cities and paid employment.

Ethnomathematics paves the way for social justice for those not employed in salaried positions [48]. Ethnomathematics provides links to a cultural identity which, in turn, will improve people's mathematical identity which is needed in all places—rural, remote, city, suburban, and small town.

**Funding:** This research received no external funding.

**Institutional Review Board Statement:** Not applicable.

**Informed Consent Statement:** Not applicable.

**Data Availability Statement:** Not applicable.

**Acknowledgments:** I want to acknowledge my co-researchers and participant co-researchers in the numerous projects in which I have been involved, especially my PNG colleagues and friends: Wilfred Kaleva, Rex Matang, Charly Muke, Sorenge Sondo, Vagi Bino, Patricia Paraide, many Elders, students, and staff at Universities and Teachers Colleges and schools and those now living in Australia, and my Australian First Nations colleagues and friends especially from Dubbo and the Clagues and co-researchers and friends Philip Clarkson and Cris Edmonds-Wathen.

**Conflicts of Interest:** The author declares no conflict of interest.

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
