# Peer review of "The Role of Mathematics Teacher Education in Overcoming Narrow Neocolonial Views of Mathematics"

_education, doi:10.3390/educsci13090868_

Round 1

Reviewer 1 Report

Well-written piece. The author seems to have extensive knowledge and experience with the education system of PNG. If I am not wrong, I feel the author has identified herself as Kay Owens. This is not appropriate for the peer review process. 

Author Response

no comment

Reviewer 2 Report

I enjoyed reading this manuscript very much. It is a lovely historic survey of mathematics education and mathematics teacher education in Papua New Guinea in the broader context of colonial history and subsequent impact of coloniality on Papua New Guinea. In general, I have few substantive recommendations. I strongly recommend publication of the eventual final version of this manuscript. However, I do feel there are ways that the manuscript could/should be improved. And I have numerous suggestions throughout the manuscript for editing and improvement.

Please read my attached document for my recommendations and comments.

I find the English language in this manuscript mostly wonderful. However, I have indicated in my attached file a number of specific paragraphs and sentences that need editing.

Author Response

see attached which is the same as I put in response to editors as I didn't realise I had to do each reviewer separately.

Reviewer 3 Report

General comments

This is an interesting paper, in the scope of Ethnomathematics, that addresses a pertinent issue for teacher education, that it is the importance of recognition of the sociocultural mathematics backgrounds of students during their formal learning. The manuscript is well written and structured, and is coherent across its different components.

The Abstract clearly and completely shows the necessary information that succinctly describes what will be developed throughout the text, from the relevance of the study, to its purpose, identifying the sources of data used as well as their analysis and the main conclusion to be drawn.  

The Research Aim & Methodology, here the purpose of the research carried out is clarified as well for the choice for a historical methodology involving a wide range of data sources. 

The Results, in this item are summary presented focusing in the “mathematics language” used in both village and schools. There is a very explicit reference, for instance to, number, counting, measurement and in particular spatial relations and visualization, in their specific language and representations used in real situations by people who do not know they are using mathematics.

If it is possible, it would be interesting to show some images of this society solving some of the problem situations of real life, described, that appeal to the use of "mathematics".

The Discussion, this is a rich item in the discussion that is made and very well grounded both in different documents, authors of reference and empirical data collected. Here I missed also, of seeing some more detailed descriptions with pictures/figures/photos of problem situations and how they were solved. It would also be interesting to present some situations in which the students, having this ancestral knowledge, transferred it to the classroom, and to know if they managed to do it or if the teacher helped them.

The Conclusion, author(s) present some ideas for teacher education and its role to sensitize both preservice and inservice teachers  to their ancestral approaches to mathematics, and the importance of ethnomathematics in dealing with this issue, and it should therefore be compulsory in teacher training so that it can provide links to a cultural identity.

Congratulations to the authors for this article which I really enjoyed reading.

Author Response

The example and application are now given.